# Normalised Precision at Fixed Recall for Evaluating TAR

## ABSTRACT

A popular approach to High-Recall Information Retrieval (HRIR) is Technology-Assisted Review (TAR), which uses information retrieval and machine learning techniques to aid the review of large document collections. TAR systems are commonly used in legal eDiscovery and medical systematic literature reviews. Successful TAR systems are able to find the majority of relevant documents using the least number of manual assessments. Previous work typically evaluated TAR models retrospectively, assuming that the system achieves a specific, fixed Recall level first and then measuring model quality (for instance, work saved at r% Recall).

This paper presents an analysis of one of such measures: *Precision at r% Recall (P@r%)*. We show that minimum Precision at r% scores depends on the dataset, and therefore, this measure should not be used for evaluation across topics or datasets. We propose its min-max normalised version ($nP@r\%$), and show that it is equal to a product of TNR and Precision scores. Our analysis shows that $nP@r\%$ is least correlated with the percentage of relevant documents in the dataset and can be used to focus on additional aspects of the TAR tasks that are not captured with current measures. Finally, we introduce a variation of $nP@r\%$, that is a geometric mean of TNR and Precision, preserving the properties of $nP@r\%$ and having a lower coefficient of variation.

## CCS CONCEPTS

• **Information systems → Evaluation of retrieval results**; *Retrieval effectiveness.*

## KEYWORDS

TAR, citation screening, evaluation, precision at recall

**ACM Reference Format:**
Anonymous Author(s). 2024. Normalised Precision at Fixed Recall for Evaluating TAR. In *Woodstock '18: ACM Symposium on Neural Gaze Detection, June 03–05, 2018, Woodstock, NY*. ACM, New York, NY, USA, 7 pages. https://doi.org/10.1145/nnnnnnn.nnnnnnn

## 1 INTRODUCTION

High-Recall Information Retrieval (HRIR) focuses on identifying nearly all relevant documents within a given collection. Technology-Assisted Review (TAR) is a prevalent method in HRIR that combines information retrieval and machine learning techniques to enhance the review of large document sets. The primary objective of TAR is to augment human effort by automating routine tasks and prioritising documents for review, thereby saving time and resources for organisations.

Citation screening for systematic literature reviews is a key application of TAR [20, 22, 23, 32]. In this task, researchers screen a large number of publications initially identified through a literature search to determine those relevant to the review. This process, traditionally manual, is time-consuming and demands extensive effort, involving numerous eligibility decisions. Other TAR applications include legal electronic discovery [11, 40] or constructing evaluation collections [28]. Initiatives such as TREC Legal [2, 11, 31, 36], TREC Total Recall [16], and CLEF eHealth TAR [20–22], have facilitated HRIR research by providing datasets and standardised evaluation methods.

A critical metric for HRIR systems is Recall, indicating the fraction of relevant documents retrieved. TAR aims to maximise relevant document identification (True Positives, *TP*) while minimising the inclusion of irrelevant ones (False Positives, *FP*). By decreasing *FP* counts, TAR systems enhance efficiency for reviewers. Nonetheless, implementing TAR requires care, as subpar performance can lead to legal repercussions, personal liability, and financial losses, especially in legal discovery contexts [14].

Various evaluation measures have been proposed to assess the effectiveness of TAR systems [37]. One prevalent approach is to evaluate the system at a fixed Recall level. This approach has been popularised by methods measuring work saved compared to the random ordering of documents (e.g., Work Saved over Sampling, *WSS@r%* [9]) and by counting True Negatives at an r% Recall (*TNR@r%*) [25]. Evaluating TAR systems at a fixed Recall level aids in determining the trade-off between Precision and Recall. Traditionally, this has been particularly useful under the assumption that a minimum acceptable level of Recall exists for a task.

As the number of potential applications for TAR grows, so too does the need for enhanced evaluation techniques. In this paper, we examine one of the measures used for evaluating TAR systems: Precision at r% Recall (*Precision@r%*, *P@r%*) [23, 26]. We find that it does not fulfil the zero Axiom #3 introduced by Busin and Mizzaro [5]. To address this limitation, following the approach of the nDCG measure [19], we propose to min-max normalise it.

Our contributions are as follows:

- We analyse the Precision at r% Recall measure and propose a min-max normalised Precision at r% ($nP@r\%$), equating to the product of $P@r\%$ and $TNR@r\%$.
- We conduct experiments to investigate the differences in evaluations and rankings using $nP@r\%$ compared to other TAR metrics. We show that $nP@r\%$ is the least correlated with the percentage of relevant documents in datasets among considered metrics.
- We introduce $snP@r\%$, a geometric mean of $TNR$ and Precision, preserving the properties of $nP@r\%$ and having lower coefficient of variation.

We first briefly describe Technology-Assisted Reviews. Then, we propose an analytical formulation of normalised Precision at a Recall rate. Finally, we conduct experiments to compare $nP@r\%$ and $snP@r\%$ with other popular TAR measures. The source code for our experiments is publicly available.[1]

## 2 BACKGROUND

All TAR automation models can be coarsely categorised into prioritisation (ranking) or classification approaches [32]. An effective TAR algorithm aims to maximise the number of relevant documents found and save the reviewers' time by removing irrelevant documents.

When treating the TAR as a ranking task (e.g., for the sub-task of screening prioritisation or stopping prediction), then rank-based measures and measures at a fixed cut-off are commonly used, e.g., $nDCG@n$, $Precision@n$, $Recall@n$, R-Precision [16], and last relevant found.

When TAR is treated as a classification task, measures based on the confusion matrix and the notion of Precision and Recall are commonly used [32, 37]. Aside from Precision and Recall, measures include variations of the harmonised mean between the two, i.e., $F_\beta$–score, Yield, Burden [39], $Utility_\beta$ [38], sensitivity-maximising thresholds [12], and AUC [8]. Another measure, Work Saved over Sampling ($WSS$), measures the amount of work saved when using machine learning models to screen irrelevant publications [9]. The True Negative Rate ($TNR$) was proposed as an alternative as it addresses some of the limitations of $WSS$ regarding averaging scores from multiple datasets [25]. Retrospectively evaluating models at different levels of Recall takes into account the number of relevant documents found and the trade-off between reviewing more documents and potentially finding more relevant ones, versus stopping the review and potentially missing some relevant documents.

Recall versus effort plots using the *knee method* [10] have been proposed as a more generalised extension, plotting the scores over the full range of values of Recall. However, these methods, similarly to the ROC curve do not provide users with a single number score, which might be crucial for some users.

Yang et al. [41] proposed a mathematical model that predicts how varying document and reviewer costs affect total TAR workflows. However this framework focuses on cost modelling for reviewing one specific query.

Previous work used Precision at r% Recall as an evaluation measure for automated citation screening algorithms [23, 24]:

$$Precision@r\% = P@r\% = \frac{TP}{TP + FP}, \text{ when } Recall = r\% \quad (1)$$

Researchers used $P@r\%$ to evaluate also other tasks like classification [7, 30, 33] or object detection [17]. Another application was mining user query logs to refine component description [29]. In the medical and healthcare domains, $P@r\%$ is referred to as "PPV at sensitivity level" and has been used for evaluation in several other works, such as in [3, 4, 6, 13, 18].

Consider an example scenario in which a search for a review returns a collection of $N = 2,000$ documents. Of these, 200 are relevant to the study and should be included in the final review (we

[1]https://anonymous.4open.science/r/normalised-precision-at-recall-D246

call these ground truth relevant items *includes*, $\mathcal{I} = TP + FN$), while the remaining 1, 800 are irrelevant and should be excluded (we call them *excludes*, $\mathcal{E} = TN + FP$ ). In manual screening, annotators must review all 2, 000 documents to identify only the 200 relevant ones. In the case of TAR systems, we consider that some of these irrelevant documents will be correctly identified by the model.

The domain and characteristics of the review influence the choice of Recall level. Past studies on the automation of citation screening in medicine typically used 95% Recall as the threshold to preserve a satisfactory quality of the systematic literature review in medicine [9]. In other technology-assisted review domains, Recall levels might be lower, for instance, in eDiscovery, a commonly used Recall is 80% [40, 42]. Sometimes the choice of Recall is influenced by the time or money limitations of the task.

## 3 NORMALISED PRECISION AT $r\%$ RECALL

Defining a Recall level for assessing TAR systems assumes that the number of true positive and false negative documents remains constant. Achieving a specific $r\%$ Recall assumes that exactly $(1 - r)\%$ of documents that should be included will be misclassified. Therefore, for a specific $r\%$ Recall, the number of True Positives ($TP$) and False Negatives ($FN$) will be equal to:

$$TP = r \cdot |\mathcal{I}|, \quad (2)$$

$$FN = (1 - r) \cdot |\mathcal{I}|. \quad (3)$$

This means that these terms will also be a constant for every model for the same dataset. For instance, from the example in the previous section, a Recall of 95% is achieved when the model accurately identifies 190 relevant documents ($TP$) and misclassifies the remaining 10, i.e., these are False Negatives ($FN$). The Precision of the model depends on the number of False Positives ($FP$), which can range from zero (best score) to the number of all excludes ($|E|$, worst score). Using the above equations, we can define maximum and minimum Precision@r% values as follows:

$$max(Precision@r\%) = \frac{r \cdot |\mathcal{I}|}{r \cdot |\mathcal{I}| + 0} = 1, \quad (4)$$

$$min(Precision@r\%) = \frac{r \cdot |\mathcal{I}|}{r \cdot |\mathcal{I}| + |\mathcal{E}|}. \quad (5)$$

Maximum Precision@r% value will always be equal to 1. However, the minimum Precision value, similarly to WSS measure [25], depends on the $\mathcal{I}/\mathcal{E}$ ratio of the dataset:

$$\lim_{|\mathcal{E}| \to 0} min(Precision@r\%) = \lim_{|\mathcal{E}| \to 0} \frac{r \cdot |\mathcal{I}|}{r \cdot |\mathcal{I}| + |\mathcal{E}|} = 1, \quad (6)$$

$$\lim_{|\mathcal{I}| \to 0} min(Precision@r\%) = \lim_{|\mathcal{I}| \to 0} \frac{r \cdot |\mathcal{I}|}{r \cdot |\mathcal{I}| + |\mathcal{E}|} = 0. \quad (7)$$

For datasets highly imbalanced towards the negative class, the minimum value of P@r% will be close to 0. On the other hand, with a growing presence of the positive class, the minimum value of P@r% will be growing towards 1.

Busin and Mizzaro [5] introduced an axiomatic approach to IR evaluation measures proposing eight axioms that every effectiveness metric should satisfy. Axiom #3 (Zero and maximum) states:

"*An effectiveness metric should have a true zero in 0 and a maximum value M. The theoretically worst (best) performances $\perp$ should give 0 (M) as the metric value. As a normalisation convention let M = 1 such that $\forall$ metric, $range(\text{metric}) = [0, 1]$, $\text{metric}(\alpha, \alpha) = 1$, and $\text{metric}(\alpha, \perp) = 0$.*"

The minimum Precision value, depending on the class imbalance, violates the aforementioned Axiom #3. This becomes crucial, especially in retrieval tasks, where the scores are almost always averaged across several topics or datasets. $P@r\%$ is favouring those models underperforming on easier topics, which consequently narrows the gap between good and poor models. Therefore, we argue that this measure should not be employed for such evaluations. To address this problem and facilitate averaging across datasets, we propose defining a min-max normalised version of $Precision@r\%$ ($nP@r\%$):

$$nP@r\% = \frac{\frac{TP}{TP+FP} - \frac{TP}{TP+|\mathcal{E}|}}{1 - \frac{TP}{TP+|\mathcal{E}|}}$$

$$nP@r\% = \frac{\Big(TP \cdot (TP + |\mathcal{E}|) - TP \cdot (TP + FP)\Big)/\Big((TP + FP) \cdot (TP + |\mathcal{E}|)\Big)}{\Big(\cancel{TP} + |\mathcal{E}| - \cancel{TP}\Big)/\Big(TP + |\mathcal{E}|\Big)}$$

$$nP@r\% = \frac{TP \cdot |\mathcal{E}| - TP \cdot FP}{(TP + FP) \cdot \cancel{(TP + |\mathcal{E}|)}} \cdot \frac{\cancel{(TP + |\mathcal{E}|)}}{|\mathcal{E}|}$$

$$nP@r\% = \frac{TP \cdot (|\mathcal{E}| - FP)}{(TP + FP) \cdot |\mathcal{E}|}$$

$$nP@r\% = \frac{TP \cdot TN}{(TP + FP) \cdot |\mathcal{E}|}$$

$$nP@r\% = \frac{TP \cdot TN}{(TP + FP) \cdot (TN + FP)}$$

$$nP@r\% = \frac{TP}{TP + FP} \cdot \frac{TN}{TN + FP}, \tag{8}$$

where the following equation can be resubstituted as:

$$nPrecision@r\% = \frac{TP}{TP + FP} \cdot \frac{TN}{TN + FP} = P@r\% \cdot TNR@r\% \tag{9}$$

$$nP@r\% = P@r\% \cdot TNR@r\%. \tag{10}$$

Equation (10) shows that $nP@r\%$ is interconnected with Precision and True Negative Rate. Interestingly, both measures relate to type I error ($FP$). Achieving high normalised Precision requires a balance between identifying relevant documents (Precision) and disregarding irrelevant ones (Specificity). This relationship can be important for evaluating and improving information retrieval models, especially in contexts of high-recall search tasks.

As Precision scores tend to have high variance in comparison with other measures, we propose to further introduce a variation of the $nP@r\%$ which is a geometric mean of its components:

$$snP@r\% = \sqrt{nP@r\%} = \sqrt{P@r\% \cdot TNR@r\%} \tag{11}$$

By introducing the square root, we intend to decrease the influence of Precision. The formulation in Equation (11) is analogous to the Fowlkes–Mallows index [15], a clustering similarity measure, where the $TPR$ term would be replaced with $TNR$. $snP@r\%$ also preserves the zero Axiom.

## 4 EXPERIMENT SETUP

To assess the importance of our findings, we conduct experiments comparing $nP@r\%$ scores (also abbreviated as $nP$ in subsequent sections) with other measures. We select the task of ranking documents for a systematic review search. We conduct the experiments using 100 systematic reviews (topics) from the CSMeD-Cochrane-dev benchmark [27]. CSMeD-Cochrane is a meta-dataset combining five different test collections [1, 20–22, 35]. CSMeD-Cochrane is the most extensive collection of systematic reviews used to evaluate document screening algorithms.

We select this dataset due to its extensive coverage of topics and public availability of baseline runs.[2] We reuse runs described in the original CSMeD paper, which includes five different models: two statistical models (BM25 and TF-IDF), and three Transformer-based models (MiniLM-L6-v2[3], MPNet-base-v2[4] and BioBERT-snli[5]) from the SentenceTransformers library [34]. Each of the five models uses four different systematic review meta-data as input query representations: 'title', 'abstract', 'eligibility criteria' and 'search strategy'. This configuration results in a total of 20 different combinations of runs.

We re-evaluate the runs at the Recall level of 95%, using $nP$, $snP$ and the two measures that are part of the equation: *Precision* and $TNR$. We also calculate other standard TAR evaluation measures: Mean Average Precision ($MAP$) and average position at which the last relevant item is found calculated as a percentage of the dataset size ($LastRel$) [20]. We intentionally refrain from using $WSS$ measure as previous work highlighted its limitations and demonstrated that $WSS$ is a special version of $TNR$ [25].

## 5 RESULTS AND DISCUSSION

We first look at correlation between $nP$ and other measures. Then we investigate the change in run rankings for each measure and finally we evaluate the impact of different levels of Recall.

### 5.1 Correlation between measures

Table 1 presents correlations between measures using Spearman's method. There is a moderate correlation between $nP@95\%$ (and $snP@95\%$) and all other measures (between .655 and .533). Especially between $P@95\%$ and $TNR@95\%$ correlations are comparable meaning a comparable influence of both components of the equation. Interestingly, $nP@95\%$ (and $snP@95\%$) exhibits the weakest, almost negligible, correlation between percentage of relevant examples, in contrast to all other considered measures. We also measure a correlation with a dataset size defined as a total number of documents found by a search query ($|\mathcal{E}| + |\mathcal{I}|$). We find that $nP@95\%$ shows a weaker correlation to dataset size when compared to MAP. This difference highlights that *nPrecision* focuses on distinct aspects of the screening task. Detailed plots presenting correlations between $nP@95\%$ and $P@95\%$ and $TNR@95\%$ are in Appendix A.

Figure 1 presents presents the coefficient of variation (CV) in evaluation measure scores between topics as depicted through violin plots for normalised measures. $nP@95\%$ shows a high variance

---

[2]Available from https://github.com/WojciechKusa/CSMeD-baselines
[3]https://huggingface.co/sentence-transformers/all-MiniLM-L6-v2
[4]https://huggingface.co/sentence-transformers/all-mpnet-base-v2
[5]https://huggingface.co/pritamdeka/S-BioBert-snli-multinli-stsb

**Table 1: Correlation matrix of selected metrics calculated using Spearman's method. $nP@95\%$ and $snP@95\%$ have identical correlation coefficients.**

|  | $nP@95\%$ $snP@95\%$ | P@95% | TNR@95% | LastRel | MAP |
|---|---|---|---|---|---|
| P@95% | 0.602 | 1. | -0.027 | 0.140 | 0.910 |
| TNR@95% | 0.655 | -0.027 | 1. | -0.923 | 0.014 |
| LastRel | -0.533 | 0.140 | -0.923 | 1. | 0.097 |
| MAP | 0.570 | 0.910 | 0.014 | 0.097 | 1. |
| Dataset size ($|\mathcal{E}| + |\mathcal{I}|$) | -0.299 | -0.724 | 0.273 | -0.249 | -0.637 |
| % Relevant | 0.132 | 0.736 | -0.570 | 0.652 | 0.639 |

**Table 2: Ranking of runs based on each average score for each measure for top 8 runs according to $nP@95\%$ score.**

| Run | nP@95% | snP@95% | P@95% | TNR@95% | LastRel | MAP |
|---|---|---|---|---|---|---|
| MPNet$_{abstract}$ | 1 | 1 | 1 | 1 | 1 | 1 |
| MPNet$_{criteria}$ | 2 | 2 | 2 | 2 | 2 | 2 |
| MiniLM$_{criteria}$ | 3 | 4 | 5 | 4 | 5 | 5 |
| MiniLM$_{abstract}$ | 4 | 3 | 4 | 3 | 3 | 3 |
| MPNet$_{title}$ | 5 | 5 | 3 | 5 | 4 | 8 |
| BioBert$_{criteria}$ | 6 | 7 | 7 | 6 | 6 | 4 |
| MiniLM$_{title}$ | 7 | 6 | 6 | 7 | 8 | 9 |
| BM25$_{abstract}$ | 8 | 9 | 8 | 11 | 10 | 6 |
| BioBert$_{abstract}$ | 9 | 8 | 10 | 8 | 11 | 7 |
| BM25$_{title}$ | 10 | 11 | 9 | 12 | 13 | 11 |
| ... | ... | ... | ... | ... | ... |

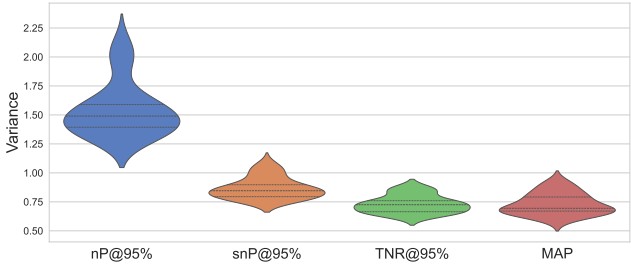

**Figure 1: Coefficient of variation in evaluation measure scores between topics presented as violin plots for normalised measures.**

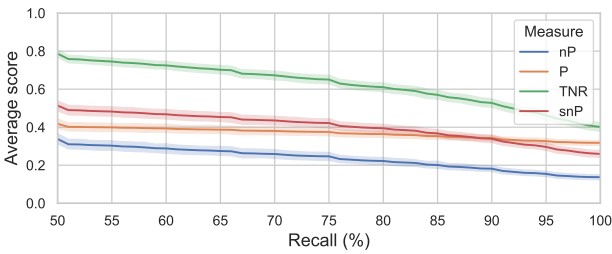

**Figure 2: Evaluation measure scores averaged across datasets and runs depending on selected Recall level.**

as their mean CV from 20 runs is equal to 1.5. This behaviour is influenced by Precision, which disproportionately favours better-performing systems. $TNR$ and $MAP$ exhibit comparably lower variances, might be considered better metrics for discriminating between good and bad systems, as they show less sensitivity to variations across different queries. However, we observe that the mean CV for $snP@95\%$ falls within the range of the mean CV for MAP, which is considered a reliable evaluation measure. This validates our assumption to use the geometric mean for reducing the impact created by the high variance in Precision.

### 5.2  Change in run ranking

We can observe that the ordering of run changes when different metrics are applied (see Table 2). Especially, $nP@r\%$ offers a different perspective for ordering when contrasted with all other measures and, especially with the incorrect usage of $P@r\%$. However, these differences are not statistically significant for the top 10 runs. We hypothesise it is due to the large collection size, which contains various topics of very different types and characteristics. Analysis on a larger number of datasets and models could enhance these findings.

### 5.3  Influence of Recall level

Figure 2 presents average evaluation measure scores across datasets and runs depending on the selected Recall level. Notably, as the Recall threshold is increased, Precision predictably diminishes due to the typical trade-off between these metrics—increasing the number of True Positives often results in a proportional increase in False Positives, thus reducing Precision.

The $nP$ measure is sensitive to changes in both Precision and TNR, and the trend in $nP$ indicates that it is likely being more heavily influenced by Precision than TNR, given the shape of its curve in relation to the other two measures. This observation underscores the utility of the $nP$ in scenarios where both False Positives and False Negatives carry significant costs.

### 6  CONCLUSION

This paper analyses Precision at $r\%$ Recall behaviour as an evaluation measures in a high-recall setting. We show the problems with using Precision@$r\%$ and propose its min-max normalised version. nPrecision at $r\%$ is equal to the product of Precision and True Negative Rate, offering a comprehensive measure for benchmarking IR systems, emphasising the need for models to optimise both True Positives and True Negatives. We also introduced $snP$, a variation of $nP$ that is the geometric mean of Precision and TNR.

We presented empirical analysis of $nP@r\%$ and compared it to other TAR measures. We showed how these evaluation measures can be used to focus on models' performance on different aspects of the screening process. Notably, $nP@r\%$ and $snP@r\%$, among all tested measures, has the lowest correlation with the percentage of relevant documents in dataset, making it more robust to evaluating screening models. For Recall-oriented tasks, high TNR is desirable but not sufficient on its own, as it does not account for the ranking of retrieved items. $nP$ and $snP$ can be important measures since they also assesses the quality of the ranking. In future work, we will focus on evaluating and estimating $snP$ scores within legal eDiscovery workflows.

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

## A    DETAILED CORRELATION PLOTS

Figure 3 presents scatter plots contrasting $nP@95\%$ with $P@95\%$ and $TNR@95\%$ scores across runs and datasets. The plot reveals a range of values for both metrics across the tested models, indicating variability in performance. The size of each marker represents the relative percentage of relevant documents in the dataset with larger markers meaning datasets with higher ratio of relevant documents. Correlations mentioned in Section 5.1 can be observed for both component measures of $nP@95\%$. For example, datasets consisting of a larger number of relevant documents (represented by larger circles) exhibit higher $P@95\%$ scores. However, this cannot be observed for $nP@95\%$.

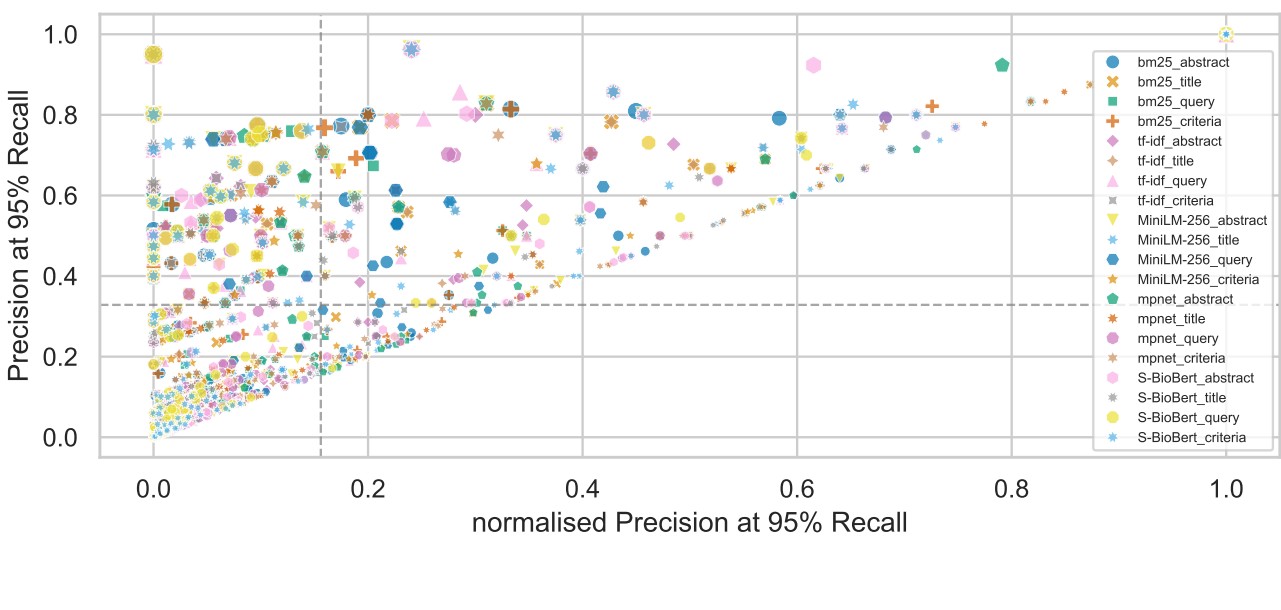

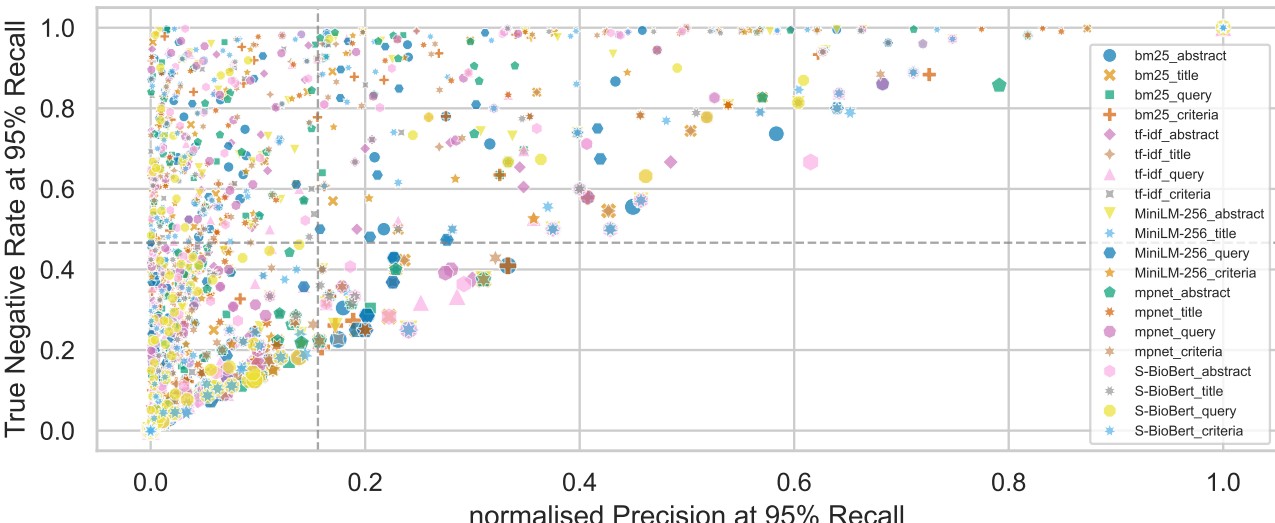

**Figure 3: Scatter plots of normalised Precision ($nP$) versus Precision (*top*) and TNR (*bottom*) at 95% Recall across twenty tested runs. Figures illustrate the trade-off between scores. The size of each marker represents the relative percentage of relevant documents in the dataset. Average $nP@95\%$, $P@95\%$ and $TNR@95\%$ are indicated by dashed grey lines.**