# OpenReview forum: "Normalised Precision at Fixed Recall for Evaluating TAR"
_ACM.org/SIGIR/ICTIR/2024/Conference — ICTIR 2024_

### Official Review · Reviewer_Ai75 · 2024-05-12

**Rating:** 2
**Confidence:** 4

**Objective Part Of Review:**

In this paper the authors observe that P@r% has a constant upper bound (1), but the lower bound is dependent on the data set because the number of non-relevant documents varies with the specific query.  They propose nP@r% and snP@r% as alternatives.  These are, in essence, the precision score linear scaled to the range 0 to 1.  The consequence is that they can be averaged over multiple queries without bias.

This reviewer has very little feedback on this paper.  Could nP@r% be used in metrics such as MAP rather than using P@n?

Typos
Page 3: presents presents -> presents

**Subjective Part Of Review:**

The paper puts a good case for using the new metric.  The derivation is appropriate.  The experiments appear good. And it is well written.

---

### Official Review · Reviewer_hQwc · 2024-05-16

**Rating:** 2
**Confidence:** 3

**Objective Part Of Review:**

The authors analyze the measure ‘precision at r% recall’ for technology-assisted review (TAR) systems, and show that scores using this measure depend on the dataset and should not be used across different topics/datasets for evaluation. Alternative versions are proposed, and some formal evaluation is provided for the new metrics.

The paper shows good results on the new metric, such as decreased variance between runs and a low correlation with the percentage of relevant documents in the dataset - a very desirable property for screening systems.

The claim made in 5.1 about the CV of snP and MAP being in the same range is maybe a bit weak, there is a clear difference but I guess it's technically correct; generally, I think it's fair to say that it is comparable variance ct. MAP, and certainly better than nP in these examples.

The writing is clear and concise, and results are presented in an easy to understand manner. Theoretical foundation behind the proposed changes is a great addition to the paper.

Small remarks: overfull hbox in equation 8

**Subjective Part Of Review:**

I am not very familiar with the specific topic "TAR systems" of the paper, but all presented evidence and literature looks reasonable to me. The presented metrics, correlations etc. are convincing to me. The problem is described clearly and the proposed solution seems to alleviate it, so I think this is a good contribution - perhaps for a subsection of the conference attendees, but still.

---

### Official Review · Reviewer_HPCB · 2024-05-17

**Rating:** 2
**Confidence:** 4

**Objective Part Of Review:**

The problem is clearly stated, and the work is easy to follow. The literature review looks exhaustive and seem to cover the important parts.

The claims are supported, as the new measure has a true 0. We might, however, also look at extreme cases – what if there are no non-relevant documents, in which nP@r% would result in a division by zero?

All in all, well-written and everything looks fine.

**Subjective Part Of Review:**

The paper was easy to read and understand and the contribution was made clear early on. It makes a nice theoretical contribution and is therefore very much in the scope of ICTIR. I only wished for some more discussion at times (see my official comments).

---

### Official Review · Reviewer_PUjr · 2024-05-21

**Rating:** 0
**Confidence:** 4

**Objective Part Of Review:**

Section 3 clearly state the problem of calculating the average of precision@r measures over different test datasets for performance evaluation of different TAR methods / systems. This section also introduce the proposed normalization and how it corrects this issue. Pointing out this problem with averaging different precision@r measure over different datasets is important for a fair and objective comparison of different TAR-Methods/systems.

Section 4 runs some experiments with the proposed new metric and compares it again existing metrics. This section is descriptive, but the derived insights are limited (e.g., change in run ranking "However, these differences are not statistically significant for the top 10 runs").

There is an inconsistency in the paper:

section 2, top right column “includes” and “excludes” are defined. Here I and E are used instead of |I| and |E| (as in e.g. in eq (2) and (3)).

**Subjective Part Of Review:**

To sum-up the paper, the proposal is to normalize the precision@r measure to 0 to 1. Some descriptive experimental results are given for this new measure, which are not very insightful. At the end, I do not know whether the proposed normalization really makes a difference in comparing different TAR methods.

Also, the related work lists many performance measured without clear definitions. It would help the readability of the paper to also define most of these mentioned metrics, at least those that are used in the experimental results section.

Finally, I must confess that I am not an expert at all in HRIR. Maybe, I am underestimating the importance of the work for this field.

---

### Meta-Review · Area_Chair_N9Wa · 2024-05-26

**Recommendation:** Accept (Oral)
**Confidence:** 5

**Metareview:**

The authors proposed normalized versions of the precision@recall measure for recall-oriented tasks, such as TAR.
The reviewers' feedback is mostly positive. They noted that the paper is easy to follow and that the proposed metrics are intuitive and clearly explained. The reviewers agreed that the paper makes a small yet interesting theoretical contribution.
However, the reviews also requested some improvements to the presentation of the paper and additional clarifications and discussion, including:

- Discussing edge cases when there are no non-relevant documents.
- Providing additional explanations when discussing Equation 8.
- Correcting some typos, grammatical errors, and notation issues.

The authors should address these issues when preparing the final version.